# Oestrogen Receptor Isoforms May Represent a Therapeutic Target in Oesophageal Adenocarcinoma

**DOI:** 10.3390/cancers14081891

**Published:** 2022-04-08

**Authors:** Steven L. Due, David I. Watson, Isabell Bastian, Ann-Kathrin Eichelmann, Damian J. Hussey

**Affiliations:** 1Department of Surgery, Flinders Medical Centre, Bedford Park, SA 5042, Australia; steven.due@flinders.edu.au (S.L.D.); david.watson@flinders.edu.au (D.I.W.); 2Flinders Health and Medical Research Institute—Cancer Program, College of Medicine and Public Health, Flinders University, Bedford Park, SA 5042, Australia; bell.bastian@hotmail.com (I.B.); ann-kathrin.eichelmann@ukmuenster.de (A.-K.E.)

**Keywords:** oesophageal neoplasms, receptors, oestrogen, oestrogen receptor modulators, tumour suppressor protein p53, blotting, western, flow cytometry

## Abstract

**Simple Summary:**

Oesophageal adenocarcinoma is a lethal malignancy with limited treatment options. Recent studies have identified oestrogen receptors (ERs) in this cancer, which could represent a new target for therapy. In this study, we used laboratory models of oesophageal adenocarcinoma to look for the presence of variant forms of ERs. We also assessed the response to treatment with a drug that acts through these ERs. We found that variant forms of ERs do exist in this malignancy and that some of the variants appear to be important in order for the cells to respond to treatment. This could be due to interactions between different ERs, or between ERs and other molecules that are known to be important in cancer growth. Our findings are encouraging in that drugs that act through ERs might be useful for patients with oesophageal adenocarcinoma in the future.

**Abstract:**

Oesophageal adenocarcinoma is a rapidly increasing problem in which treatment options are limited. Previous studies have shown that oesophageal adenocarcinoma cells and tissues express oestrogen receptors (ERs) and show growth suppression and apoptosis in response to ER modulator agents such as tamoxifen. ERs are known to be expressed in a number of isoforms that act together to regulate cell growth and cell death. In this study, we used western blotting to profile the expression of ERα and ERβ isoforms, and expression of the oncologically related molecules p53, HER2, and EGFR, in a panel of oesophageal adenocarcinoma cell lines. The cytotoxicity of tamoxifen in the cell lines was determined with Annexin V-FITC flow cytometry, and correlations between cytotoxicity and receptor expression were assessed using Spearman’s rank-order correlation. Oesophageal adenocarcinoma cell lines showed varying cytotoxicity in response to tamoxifen. The ER species ERα90, ERα50, and ERα46, as well as p53, were positively associated with a cytotoxic response. Conversely, ERα74, ERα70, and ERβ54 were associated with a lack of cytotoxic response. The ER species detected in oesophageal adenocarcinoma cells may work together to confer sensitivity to ER modulators in this disease, which could open up a new avenue for therapy in selected patients.

## 1. Introduction

Few malignancies carry as poor a prognosis as oesophageal adenocarcinoma. Seventy-four percent of patients die within 12 months of diagnosis [1]. The incidence of this cancer has increased six-fold in the Western world over recent decades, and this trajectory continues to rise in industrialised nations [2,3,4]. Due to the morbidity of surgery, and the late presentation of many patients with advanced disease, only 25–33% of patients undergo surgical resection [5,6]. Treatment options include definitive chemoradiotherapy with curative intent, surgery (usually preceded by neoadjuvant chemoradiotherapy), or palliative measures. Indeed, for surgical candidates with locally advanced disease, current guidelines recommend neoadjuvant therapy for all fit patients [7]. Chemotherapy, therefore, has a prominent role in patient management, though novel approaches are limited.

Oesophageal adenocarcinoma demonstrates a pronounced male predilection. One hypothesis is that female sex hormones may confer a degree of protection against the disease. A recent review summarises epidemiological data that demonstrate a lower incidence of oesophageal adenocarcinoma in women with a history of exogenous oestrogen exposure (e.g., in the form of the oral contraceptive pill or hormone replacement therapy) compared to those with no such history [8]. However, a protective effect of the oestrogen receptor (ER) modulator tamoxifen has not been demonstrated. A large Swedish cohort study assessing the incidence of oesophageal adenocarcinoma among female breast cancer patients exposed to tamoxifen did not find a statistically significant change in the subsequent risk of oesophageal adenocarcinoma [9]. Thus, oestrogen pathways may influence disease development but their role in treatment has not yet been established.

Oestrogen receptors have been detected in oesophageal adenocarcinoma biopsy tissue and resection specimens [10,11,12,13,14]. ERs are expressed in oesophageal adenocarcinoma cell lines OE-19 and OE-33, and experimental evidence demonstrates that these cell lines respond to treatment with tamoxifen [15]. These studies raise the possibility that ERs may represent a therapeutic target. In oesophageal adenocarcinoma, tamoxifen is likely to be used in combination with other agents, and previous work has suggested that tamoxifen augments the cytotoxic activity of cisplatin and 5-fluorouracil in laboratory models [14]. ER modulator therapy may therefore be useful to complement existing treatment strategies.

Tamoxifen is an interesting drug; in that it acts as a modulator of ER signalling and not simply an antagonist of the hormone oestrogen. Tamoxifen exerts a variety of effects through the activation or inhibition of ER pathways depending on local cell or tissue context. ERs are expressed as two subtypes, ERα and ERβ, which govern many aspects of tissue proliferation and cell death [8]. Furthermore, each of these ERα and ERβ subtypes may be expressed in a variety of isoforms, which arise from the use of alternative promotor sites in gene transcription, alternative RNA splicing during translation, and post-translational modification [16]. These isoforms mediate a variety of contrasting activities in response to tamoxifen. Thus, it is important to understand expression patterns of ER isoforms in oesophageal adenocarcinoma cells, which may underpin response patterns in this disease.

In addition to the interaction between different isoforms, ERs also exhibit crosstalk with signalling networks that are relevant to oncogenesis, in particular, p53, EGFR, and HER2. In breast cancer, ERα binds to p53 and inhibits its activity in the DNA damage repair pathway, promoting the growth and survival of neoplastic cells and facilitating cancer progression [17]. Additionally, EGFR and HER2 interact with the nongenomic signalling pathways of ER; they regulate the expression of ERα isoforms and play major roles in the development of tamoxifen resistance in breast cancer by providing alternative pathways for cancer growth [18]. Given that p53 mutation and EGFR and HER2 overexpression are common occurrences in oesophageal adenocarcinoma, and that ERs interact with all three of these molecules, understanding this relationship may have implications for treatment response.

In this study, we have assessed ERα and ERβ expression in a number of oesophageal adenocarcinoma cell lines, in order to ascertain which ER species are expressed, together with the expression of p53, EGFR, and HER2. We also assess the cytotoxic activity of the tamoxifen metabolite 4-hydroxytamoxifen in these cell lines. Our purpose in this is to understand whether particular ER isoforms are associated with sensitivity or resistance to tamoxifen in oesophageal adenocarcinoma, as they are in other cancers. 

## 2. Materials and Methods

### 2.1. Cell Culture

Eight oesophageal adenocarcinoma cell lines (Eso-26, Eso-51, Flo-1, JH-EsoAd1, OACP4C, OE-19, OE-33, and SK-GT-4) were used. OE19 and OE33 were purchased from Sigma ECACC (Sigma-Aldrich, Castle Hill, Australia), the remaining oesophageal cell lines were kindly provided by Dr Nicholas Clemons from the Sir Peter MacCallum Department of Oncology, University of Melbourne, Australia. The breast cancer cell line MCF-7 was a gift from Dr Robyn Meech from the College of Medicine and Public Health, Flinders University, Adelaide, Australia. Characteristics of cell lines are summarised in Appendix A.

Cells were cultured in Roswell Park Memorial Institute (RPMI) 1640 medium supplemented with 10% heat-inactivated foetal bovine serum (FBS) supplemented with penicillin (100 units/mL), streptomycin (100 μg/mL), and Normocin (100 μg/mL). Cultures were maintained in a humidified environment at 37 °C with 5% CO_2_ and subcultured when they reached 60–80% confluence. For cell harvest, the spent medium was aspirated and cultures were washed with phosphate-buffered saline (PBS) prior to incubation with 0.5% trypsin-EDTA to facilitate cell detachment. In the semi-adherent (Eso-26) and anchorage-independent (Eso-51) cell lines, floating cells in suspension were collected prior to harvest of any adherent cells by trypsinisation. Complete medium was added in at least a two-fold excess to deactivate the trypsin and the cell suspension was centrifuged at 1500× *g* for 5 min. The supernatant was aspirated and cells resuspended in 10 mL of culture medium. An aliquot of the cell suspension was stained with trypan blue to identify nonviable cells and cells counted in a haemocytometer. Generally, 2–4 × 10^6^ cells were subcultured per 75 cm^2^ flask. Cell culture media and reagents were purchased from Life Technologies (Waltham, MA, USA). Because of the potential oestrogenic effects of phenol red and its possible interference with MTS assays, all culture media were phenol-red-free [19].

### 2.2. Treatment Experiments

For experiments, each cell line was cultured to 70% confluence, photographed, harvested, and counted. All experiments used RPMI medium supplemented with charcoal-stripped foetal bovine serum (CSS), to avoid the potentially confounding effects of steroid hormones contained within regular FBS [20]. A concentration of 10 μM 4-hydroxytamoxifen was selected for the apoptosis assay as it was estimated to provide a range of responses in oesophageal adenocarcinoma cell lines. This concentration is within the range of IC_50_ values calculated for three oesophageal adenocarcinoma cell lines in preliminary experiments (11.2 μM for OE-19, 7.2 μM for OE-33, and 9.5 μM for Flo-1), and within the cytotoxic range reported in the literature [14,21,22,23,24].

Cells were seeded simultaneously in 96-well plates (for MTS viability assay), 12-well plates (for flow cytometry apoptosis assay), and 25 cm^2^ flasks (for protein analysis). Seeding densities were 4000 cells/well in 89 μL medium in 96-well plates, and 44,700 cells/well in 1 mL medium 12-well plates. Cells were plated in six technical replicates in 96-well plates for the MTS viability assay and in three technical replicates in 12-well plates for the apoptosis assay. After 24 h, drug (or vehicle control) was added to give a final drug concentration of 10 μM per well, final vehicle concentration 0.1% ethanol, and 100 μL/well in 96-well plates and 1.12 mL/well in 12-well plates. Cells were incubated for 48 h prior to harvest for viability and apoptosis assays. As with the drug concentration used, the drug exposure time of 48 h was determined by preliminary experiments assessing cell growth characteristics and different exposure times.

At the time of seeding the plates, a 25 cm^2^ flask of each cell line was seeded in stripped medium at a density and concentration equivalent to 4000 cells/well in 96-well plates (294,100 cells/well in 6.53 mL medium per flask). This was harvested after 24 h, that is, at the time at which the treatments were to be added to the 12- and 96-well plates, so that a matched Western blot could be performed to assess protein expression at exactly the time of treatment. This was done to permit comparisons between receptor expression and treatment response.

### 2.3. MTS Viability Assay

After the treatment duration of 48 h, 20 μL of MTS reagent (333 μg/mL MTS with 25 μM phenazine methosulphate) was added to each well of the 96-well plates, and plates were incubated for 2 h. Optical density at 490 nm was measured on a Bio-Rad plate reader; mean background optical density was subtracted from the readings in each well prior to analysis.

### 2.4. Apoptosis Assay

Flow cytometry binding buffer (25 mM HEPES, NaOH, pH 7.4, 140 mM NaCl, and 1 mM CaCl_2_), annexin V-FITC (0.15 mg/mL), and propidium iodide (50 μg/mL) were purchased from Abcam (Melbourne, Victoria, Australia). Cells were harvested from 12 well plates for flow cytometry as follows. Cell medium (containing floating cells) was collected in 10 mL tubes; wells were washed with 1 mL PBS which was also collected; 500 μL trypsin was added to each well, and plates were incubated for the briefest possible time to allow for cell detachment (typically 2–5 min, depending on cell line and treatment condition). Cells were closely monitored during trypsinisation to ensure this step was as brief as possible in order to avoid enzymatic exteriorisation of phosphatidylserine and potential membrane injury, both of which can adversely affect the assay [25]. Trypsin was inactivated with 1 mL of complete medium, which was collected, and wells were rinsed with a final 1 mL of medium to ensure the collection of any remaining cells. Tubes were centrifuged at 1500× *g* for 5 min, the supernatant was aspirated and the cell pellet resuspended in 250 μL binding buffer. Care was taken to resuspend the cells very gently, as mechanical membrane disruption can lead to exposure of phosphatidylserine to annexin V, and therefore, to false-positive results in this type of assay [26]. Samples were transferred to FACS tubes on ice and incubated with 2.5 μL annexin V-FITC (final concentration 250 ng/mL) for 5 min in the dark. To each tube, 2.5 μL of propidium iodide (final concentration 250 ng/mL) was added immediately prior to analysis.

Flow cytometry was performed using an Accuri C6 flow cytometer and proprietary software (Becton Dickinson, MI, USA). A total of 2000 events were collected per sample. Gating was applied to the forward scatter/side scatter plot to exclude cell doublets and small particles of debris. Axis scales on the FL1 (annexin V-FITC) and FL2 (PI) plots were adjusted according to thresholds determined by calibration samples for each cell line. Calibration samples comprised untreated healthy control cells with no annexin V or PI added (to identify the viable cell quadrant); treated cells with annexin V only (to determine the early apoptosis quadrant); and treated cells with PI only (to identify the late apoptosis/necrosis quadrant). In order to compare responses between different cell lines, survival fractions were calculated, and early and late apoptosis were normalised to vehicle controls according to the following formulae:

Survival fraction (%) = 100 × % viability of replicate ÷ % mean viability of control replicates.

Early apoptosis (%) = % early apoptosis of replicate − % mean early apoptosis of control replicates.

Late apoptosis (%) = % late apoptosis of replicate − % mean late apoptosis of control replicates.

One-way ANOVA followed by Tukey’s post hoc test of multiple comparisons was used to assess the significance of differences in response between cell lines. Normalised data were compared with Spearman correlation.

## 3. Protein Analysis

Concurrently with treatment experiments, a 25 cm^2^ flask of each cell line was also seeded for the preparation of cell lysates from untreated cells at the time of treatment. These flasks were seeded from the same cultures as those used in the treatment experiments, at exactly the same density and concentration as the treatment experiments, in order to reflect cellular protein expression at the time of treatment as precisely as possible. In addition to the oesophageal adenocarcinoma cell lysates, the breast cancer cell line MCF-7 was cultured in identical conditions and harvested for a control sample. 

Buffers and reagents were purchased from Bio-Rad (Gladesville, New South Wales, Australia), except where otherwise stated. Cells were harvested and lysates prepared as follows. Cells were harvested by scraping, centrifuged at 1500× *g* for 5 min, and the supernatant aspirated. The cell pellet was resuspended in lysis buffer containing 1% (*v*/*v*) NP-40, 4 μM sodium azide, 50 mM Tris-HCl, and protease and phosphatase inhibitor cocktails, and sonicated at 4 °C in a water-cooled sonication unit. Lysates were centrifuged at 10,000× *g* for 10 min, the pellet of cellular debris discarded, and the sample stored in aliquots at −80 °C.

The protein content of lysates was quantified using an EZQ Protein Quantification Assay (Invitrogen, Mulgrave, Victoria, Australia) according to the manufacturer’s instruction. Briefly, lysates were spotted onto assay paper in triplicate along with a dilution series of an ovalbumin standard (from 2 mg/mL to 0.05 mg/mL), fixed with methanol, and imaged at 300 nm transillumination with a Bio-Rad Chemi-Doc MP imager. A standard curve was constructed from the relative intensity of the ovalbumin dilutions, and protein concentrations were calculated by interpolation using Carestream image analysis software (Carestream, Rochester, NY, USA).

Polyacrylamide gel electrophoresis was performed as follows. Per well, 20 μg protein was made up to a volume of 20 μL per well using Laemmli sample buffer and 65 mM DTT. Samples were reduced by heating to 95 °C for 2 min and allowed to cool at room temperature prior to loading the gel. Reduced samples were loaded in pre-cast Bio-Rad Criterion TGX 4–20% stain-free gels and electrophoresis was performed at 160 V for 70 to 120 min in a gel tank filled with running buffer. Precision Plus Dual protein standards (Bio-Rad) were loaded in appropriate lanes as molecular weight markers. Recombinant ERα and ERβ were diluted 1 in 1000 and also made up in the appropriate proportions to 20 μL in Laemmli sample buffer. Following electrophoresis, gels were activated under ultraviolet light for 5 min, and imaged for 5 to 10 s (ensuring no sample lane saturation) before and after transfer to PVDF membranes, to give pre- and post-transfer images of the gels. These were used to confirm the complete transfer of protein from gel to membrane.

Western blotting was performed as follows. Separated proteins were blotted onto methanol-activated 0.45 μm PVDF membranes in a Trans-Blot Turbo transfer unit (Bio-Rad) at 20 V (max 1.5 A) for 30 min. Total protein load images were obtained with a ChemiDoc MP imager (Bio-Rad) with an exposure time of 5 to 10 s, ensuring that no pixel saturation of protein bands occurred. PVDF membranes were incubated in blocking buffer for 1 h and then in primary antibody diluted to its working concentration in PVDF membrane buffer overnight at 4 °C on an agitator. Primary antibodies were HC-20 for ERα and H-150 for ERβ (Santa Cruz Biotechnology, Dallas, TX, USA). Membranes were washed thrice in membrane buffer prior to incubation with a 1 in 3000 dilution of horseradish peroxidase-conjugated secondary antibody (also in PVDF membrane buffer) for 1 h. Membranes were again washed thrice, and Amersham Enhanced Chemiluminescence (ECL) Western blotting detection reagent (GE Healthcare) was added immediately prior to development with a Fuji LAS-3000 Imager (Fujifilm Life Sciences, Tokyo, Japan) running ImageLab software (Bio-Rad). Sizing and quantification analysis was performed using ImageLab software, with sizing employing a point-to-point semi-log method in which molecular weight of species of interest is interpolated from curves generated from the molecular weight markers.

Quantification analysis was performed after normalisation to total protein load, as previously described [27]. In brief, PVDF membranes were imaged after protein transfer, identical areas through the centre of each lane from top to bottom were marked and the net intensity measured. By dividing the intensity of each lane with the mean intensity of the other lanes on the membrane, a normalisation factor was calculated. After blotting, the total intensity of each band on the blot was divided by this normalisation factor to provide a quantitative estimate.

Relative protein expression levels were calculated and correlated against response to tamoxifen (as measured by the flow cytometry apoptosis assay) with Spearman analysis. The expression of ER species was also compared with the expression of other ER species by Spearman correlation.

Lysates from the above experiments were also profiled for p53, EGFR, and HER2 expression. Lysate preparation for Western blot, and conditions for electrophoresis, were identical to the above except for the electrophoresis duration, which was 70 min. Primary antibodies were DO-1 for p53 (Santa Cruz Biotechnologies), CST #2232 for EGFR, and CST #2242 for HER2 (Cell Signaling Technology, Danvers, MA, USA). Protein quantification, normalisation to total protein load, and statistical analysis were performed as described above.

## 4. Results

### 4.1. Oestrogen Receptor Expression Patterns in Oesophageal Adenocarcinoma Cell Lines

ERα and ERβ expression was profiled in eight oesophageal adenocarcinoma cell lines. Along with the oesophageal cell lines, control lysates from the breast cancer cell line MCF-7, and samples prepared from recombinant ERα or ERβ (rERα and rERβ, respectively), were analysed. Western blots for ERα and ERβ are shown in Figure 1.

Five major ERα species were observed, with estimated molecular weights of approximately 90 kDa, 74 kDa, 70 kDa, 50 kDa, and 46 kDa. In MCF-7, an additional species at approximately 66 kDa was observed, consistent with the reported molecular weight of wild-type ERα, and with a similar estimated molecular weight to recombinant ERα. Two ERβ species were observed, resolving at approximately 59 kDa and 54 kDa in all oesophageal adenocarcinoma lines and MCF-7. Recombinant ERβ resolved as a band with a relative molecular weight of approximately 54 kDa. 

Relative expression levels of ERα and ERβ species, after normalisation to total protein load, are presented in Figure 2. Complete western blot and total protein load images are presented in Appendix A, and associated quantification data are presented in Appendix A. The cell lines with the highest expression of ERα90 (Eso-51, Flo-1, and OE-33) also showed the highest expression of ERα50 and ERα46. Additionally, the expression patterns of ERα74 and ERα70 appear to be similar to each other and mirror the expression of ERβ59 and ERβ54.

To investigate the potential relationships between ER species in oesophageal adenocarcinoma cells, relative expression was compared in a correlation matrix (Table 1). The most significant relationships were as follows: positive correlations between ERα90, ERα50, and ERα46; positive correlations between ERα74, ERα70, and ERβ54; and negative correlations between ERβ54 and ERα90, ERα50, and ERα46. There were also negative correlations between ERα90 and ERα70; ERα74 and ERα50; and ERα70 and ERα50. Thus, two groupings emerged from the analysis: one group comprising ERα90, ERα50, and ERα46; the other comprising ERα74, ERα70, and ERβ54; with the two groups generally being expressed in inverse proportion to one another. ERβ59 expression correlated positively with ERβ54, but the relationships between ERβ59 and the ERα species were not statistically significant. However, given its positive relationship with ERβ54 expression, ERβ59 could potentially be considered part of the latter group.

### 4.2. Cytotoxicity of 4-Hydroxytamoxifen in Oesophageal Adenocarcinoma Cell Lines

Tamoxifen treatment at 10 μM for 48 h led to statistically significant reductions in cell viability and concomitant increases in apoptosis in all eight cell lines. OE-33, Eso-26, and Eso-51 showed the highest levels of cytotoxicity; SK-GT-4 showed the lowest; the remaining cell lines showed a spectrum of responses within an intermediate group (Figure 3).

### 4.3. Expression of EGFR, HER2, and p53

Since ER signalling pathways are known to interact with several pathways that are highly relevant to oncogenesis, the expression of three key molecules (p53, EGFR, and HER2) was assessed in oesophageal adenocarcinoma cells. The reasons for this were (i) to examine whether there was any association between the expression of ER isoforms and the expression of these key molecules, which may imply a degree of regulatory dependence, and (ii) to assess whether there was a relationship between their expression and treatment response, since tamoxifen may interact with these pathways in certain contexts. Data represent the pooled results from three independent Western blots prepared from lysates collected during the experiments at the time treatments were applied. Representative blots are presented in Figure 4. Complete western blot and total protein load images are presented in Appendix A, and associated quantification data are presented in Appendix A.

p53 was detected in five cell lines, with the highest expression in Flo-1, intermediate expression in Eso-26, and lower expression in Eso-51, JH-EsoAd1, and OE-33. OACP4C, OE-19, and SK-GT-4 had no detecTable 53 kDa protein (at least, not according to the techniques and antibody employed here), although, interestingly, at higher exposures, OE-19 displayed a species with a molecular weight of approximately 44 kDa that reacted with the p53 antibody. EGFR was detected in all eight cell lines, with the highest relative expression in JH-EsoAd1, OE-33, and SK-GT-4. HER2 was detected at high levels in OE-19, moderate levels in Eso-26 and OE-33, and very low levels in OACP4C.

Relationships between p53, EGFR, and HER2 expression and treatment sensitivity were investigated with Spearman correlation and T-tests of high versus low expression. Scatterplots of rank comparing p53 expression with survival fraction demonstrated a notable outlier in the cell line Flo-1 (Appendix A). We reasoned that this could be because the potentially deleterious *TP53* missense mutation within this cell line might have a negative impact on treatment response (see Appendix A and refer to Section 5 for further explanation). Therefore, we repeated the analysis with the remaining cell lines. A significant negative correlation was found between p53 expression and survival fraction (ρ *=* −0.9643, *p* = 0.0028), indicating greater sensitivity in cell lines with higher p53 expression (Figure 5). No such relationship was identified for EGFR or HER2 using Spearman correlation.

T-tests on data grouped into high vs. low expression showed that p53-positive cell lines were more sensitive to 4-hydroxytamoxifen than those in which p53 was not detected (mean difference 16.5%, *p* < 0.0001), as shown in Figure 6. Conversely, higher expression of EGFR was associated with lower rates of cytotoxicity in response to 4-hydroxytamoxifen (mean difference 7.8%, *p* = 0.0291). The difference in treatment response versus HER2 expression did not meet the threshold for statistical significance (mean difference 6.8%, *p* = 0.0509). Overall, the results suggest an association between p53 and response to tamoxifen, and between EGFR and a degree of resistance to tamoxifen.

### 4.4. Relationship between ER Expression and Response to Tamoxifen

Oestrogen receptor expression data were correlated with treatment response data from the apoptosis assay by Spearman analysis. Significant correlations were found between cell survival and expression of ERα90 (ρ *=* −0.7857, *p* = 0.0480), ERα70 (ρ *=* 0.9286, *p* = 0.0067), and ERβ54 (ρ *=* 0.9643, *p* = 0.0028), as presented in Figure 7 (see Appendix A for analysis including the Flo-1 cell line). Relationships between survival and expression of ERα74 (ρ *=* 0.7413, *p* = 0.0881), ERα50 (ρ *=* −0.7500, *p* = 0.0663), and ERβ59 (ρ *=* 0.7143, *p* = 0.0881) were below the threshold for statistical significance.

## 5. Discussion

In this study, we demonstrated that the tamoxifen metabolite 4-hydroxytamoxifen is cytotoxic in several oesophageal adenocarcinoma cell lines. This response was associated with the expression of ERα and ERβ species, and with the expression of p53 and EGFR.

The association between p53 and response in oesophageal adenocarcinoma is interesting because p53 mutation is common in this disease (72–81% in clinical specimens). Although the mutations in the cell lines examined here are predicted to produce deleterious or non-functional consequences [28,29], several studies have demonstrated that TP53 mutations can still produce partially functional proteins [30,31,32,33,34]. In particular, our recent work in oesophageal adenocarcinoma has shown that knockout of mutant p53 results in diminished apoptosis in response to 4-hydroxytamoxifen [35]. In breast cancer, ERα66 binds p53 and recruits co-repressors to p53-regulated genes, inhibiting its tumour-suppressor activity [36]. 4-hydroxytamoxifen liberates p53 from this ERα66-p53 complex, restoring p53 functionality and its attendant antiproliferative and pro-apoptotic effects [17]. It is therefore possible that in oesophageal adenocarcinoma cells there is a relationship between mutant p53 and ERα, in which p53 is reactivated by 4-hydroxytamoxifen and remains at least partially functional. 

Unlike the other cell lines, Flo-1 showed no association between p53 expression and response to tamoxifen treatment, despite its relatively high p53 expression by western blot. COSMIC and IARC databases list Flo-1 as containing p53 mutated by a missense substitution, with predicted non-functional or deleterious functional consequences. The implication is that the dysfunctional p53 expressed in Flo-1 exhibits altered responses to ER modulators, potentially precluding the study of tamoxifen metabolites in this cell line. Relative p53 expression in this study (highest in Flo-1, with intermediate levels in Eso-26, JH-EsoAd1, and OE-33) is broadly in keeping with reported expression in these cell lines [37,38], though other authors have also detected low levels of p53 protein in SK-GT-4, potentially through the use of a different primary antibody [39]. The 44 kDa species detected in OE-19 using an anti-p53 antibody has also been described and is attributed to a point mutation in exon 9 of *TP53* that encodes a premature stop codon, resulting in a truncated p53 variant [37].

In the analysis of ER species and response, there are two groups of ER isoforms. The first group comprises ERα90, ERα50, and ERα46, and corresponds with tamoxifen sensitivity; the second comprises ERα74, ERα70, and ERβ54, and corresponds with tamoxifen resistance. These groups are expressed in inverse proportion to each other, implying regulatory co-dependence, which is a known feature of ER isoform function [16]. There is a well-described negative regulatory role of ERα upon ERβ, and vice versa [16,40,41]. ERβ2 (the most widely studied variant of ERβ; approximately 55.5 kDa) is known to dimerise with ERα66, inhibit its transcriptional activity, and target it for proteosomal degradation in breast cancer [42]. It is possible that a similar interaction between ERβ54 and some of the ERα species described herein may be occurring in oesophageal adenocarcinoma cells, and indeed, that ERβ54 may represent ERβ2. However, no such interaction has been demonstrated here, and these comments are confined to associations only. 

The 66 kDa ERα species detected in breast cancer cell line MCF-7 corresponds in molecular weight to the recombinant protein and is likely to represent full-length ERα. This species is not detected in oesophageal adenocarcinoma cells, though there are a number of higher- and lower-molecular weight species expressed in both oesophageal and breast cancer cells. Given the proclivity of ER isoforms to co-regulate the expression of other isoforms [16], it is postulated that the combination of ER species expressed in oesophageal adenocarcinoma cells has resulted in the suppression of full-length ERα in this disease.

The 50 kDa and 46 kDa ERα species may represent the known isoforms ERα-short (isoform 2, 53.7 kDa), ERα46 (isoform 3, 47.6 kDa), or ERα36 (isoform 4, 35.6 kDa). Post-translational modifications are relatively common in ER biology and can significantly alter electrophoretic mobility [43]. Thus, the two species may represent two distinct isoforms or one isoform and a post-translationally modified variant. ERα46 is known to mediate cytotoxic responses to 4-hydroxytamoxifen in breast cancer [8]. The relationship between the 46 kDa and 50 kDa ERα species and tamoxifen sensitivity we report is in keeping with the described role of this isoform in the literature.

The higher molecular weight ERα species expressed in MCF-7 and the oesophageal cell lines (90, 74, and 70 kDa) are less commonly described in the breast cancer literature. An 80 kDa ERα isoform has been described in the MCF-7:2A cell line, which is a clone of MCF-7 that was cultured in oestradiol-free medium for a protracted period [44,45]. Another report describes 130, 110, and 92 kDa ERα isoforms in MCF-7, detected with a C-terminally directed monoclonal antibody, and these were found to be membrane-associated receptors [46]. The higher molecular weight of these species was attributed to post-translational modifications necessary for membrane localisation, and they initiated signal transduction in response to the binding of oestrogens, suggesting that they remain functional variants of ERα. Given that ERα has been detected in both the nucleus and cytoplasm of oesophageal adenocarcinoma cells, it is possible that the 90, 74, and 70 kDa species may be ERα variants with modifications that localise the species to cytoplasm or membrane [14]. 

An alternative theory is that the higher molecular weight species may represent dimers. Dimerisation is an integral aspect of ERα and ERβ function, and dimers with resistance to typical reducing conditions have been described. Activated ER tends to dimerise in the presence of ligand, prior to translocation to the nucleus and initiation of DNA transcription at oestrogen response elements [16]. Dimerised ER also exists in the absence of ligand, and these dimers may be highly resistant to denaturation due to the strong hydrophobic forces mediating dimerisation in vivo [47,48]. Thus, ERα90 could represent a dimerised form of ERα46, which would explain both its apparent molecular weight and its association with a cytotoxic response to 4-hydroxytamoxifen.

The two ERβ species detected by western blot at approximately 59 and 54 kDa are compatible with ERβ1 (59.2 kDa) and ERβ2 (55.5 kDa), and their negative relationship with ERα expression is in keeping with the described roles of these isoforms in the breast cancer literature [41]. Equally, however, the 54 kDa species could represent one of several other ERβ variants described in humans, such as ERβ4 (54.1 kDa), or ERβ5 (53 kDa) [49]. These isoforms do not bind ligand due to rearrangements that make the ligand-binding pocket inaccessible but retain a functional dimerisation domain and can heterodimerise with ERβ1 (to enhance AF-2 activation) and with ERα (to inhibit transcriptional activity) [50].

EGFR was widely expressed in oesophageal adenocarcinoma cell lines and was associated with resistance to 4-hydroxytamoxifen. EGFR overexpression is a common feature in oesophageal adenocarcinoma, with higher expression being associated with adverse clinical outcomes and resistance to chemotherapy [51]. In breast cancer, EGFR is an alternative survival pathway that becomes activated in tamoxifen-treated cancers to mediate resistance to the drug and promote ongoing cancer progression [18]. Thus, the findings of an association between EGFR and tamoxifen-insensitivity in oesophageal adenocarcinoma cells may represent either a more aggressive tumour phenotype reflected in higher EGFR expression or the activation of an alternative pathway by which cell proliferation continues in the presence of the drug. Studies quantifying EGFR levels in oesophageal adenocarcinoma cell lines in the literature are sparse, but the limited data available are in keeping with this report of higher expression in OE-33 and SK-GT-4 and relatively low expression in OE-19 [52]. Given the potential association between EGFR overexpression and resistance to tamoxifen, future work investigating the blockade of EGFR in tandem with ER in this disease could be valuable.

We acknowledge several limitations of our study. The observations reported here are associative, and causal relationships between ER isoform expression and response have not been demonstrated. Furthermore, the ERα and ERβ species reported here have not been fully characterised. Future work to characterise the ERα and ERβ species we describe, for instance using either isoform-specific antibodies or mass spectrometry, could clarify which isoforms are expressed and address the questions above around dimer formation and post-translational modification.

An additional limitation is that much of the epidemiological data implicating oestrogens as a protective factor in oesophageal adenocarcinoma is derived from female patients, but the experiments presented here were performed in cell lines derived largely from males (7 male, 1 female; see Appendix A). Androgens and androgen receptor (AR) pathways were not investigated in this study and may represent a worthwhile line of inquiry. They have been reviewed elsewhere [53].

A further limitation is that while the expression of related molecules p53, EGFR, and HER2 was profiled at a protein level using Western blot, an analysis of their mutation status at a genetic level was not determined. As such, the influence of mutant forms of these proteins upon cell growth and response to pharmaceutical agents in the experimental models we employed is unknown.

## 6. Conclusions

To conclude, it is evident that tamoxifen is cytotoxic in oesophageal adenocarcinoma cells in vitro. Several ERα and ERβ species are expressed in these cell lines, and their expression is associated with either sensitivity or resistance to ER modulator therapy. This may occur due to interactions between different ER isoforms, and potentially via interactions between these isoforms and other oncologically related pathways such as p53, EGFR, and HER2. This suggests that ER modulators such as tamoxifen, acting via ERα and ERβ networks to induce a cytotoxic response, may have a role in the clinical treatment of patients with oesophageal adenocarcinoma. Further investigation to clarify the identity of the ER species associated with sensitivity or resistance to tamoxifen may help in developing a targeted approach to patient treatment.

## Figures and Tables

**Figure 1 cancers-14-01891-f001:**
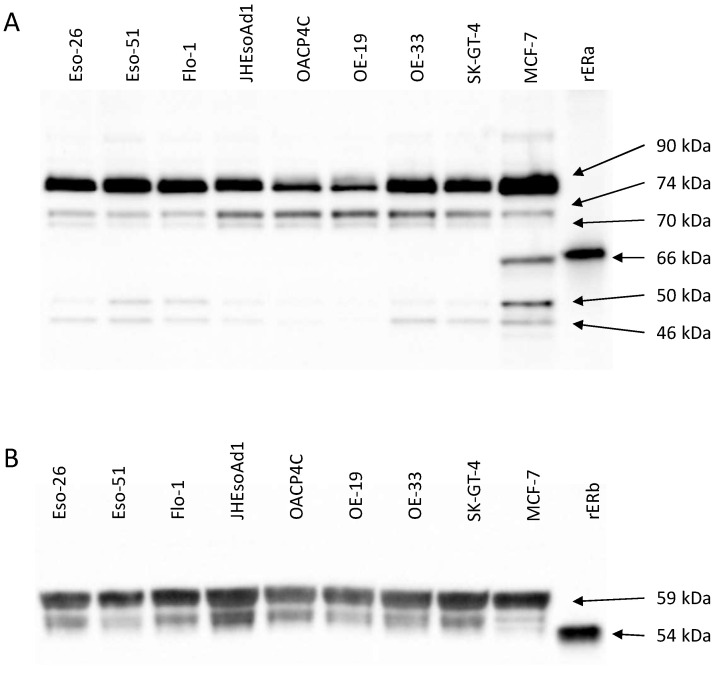
Expression of ERα (panel (**A**)) and ERβ (panel (**B**)) in oesophageal adenocarcinoma and breast cancer cells, by Western blot. rERα and rERβ, recombinant ERα and ERβ, respectively; kDa, kilodaltons.

**Figure 2 cancers-14-01891-f002:**
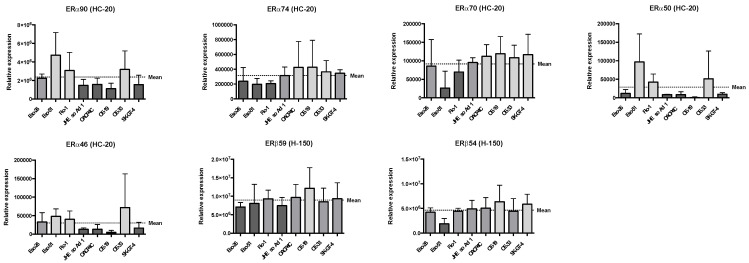
Relative expression of ERα and ERβ species by Western blot, normalised to total protein load. These graphs represent data from three independent Western blots, which were generated using lysates obtained during the three independent experiments from which apoptosis data are reported below and were harvested at the time treatments were applied to the cells. Plots represent means and error bars the standard deviation. HC-20, anti-ERα primary antibody; H-150, the anti-ERβ primary antibody.

**Figure 3 cancers-14-01891-f003:**
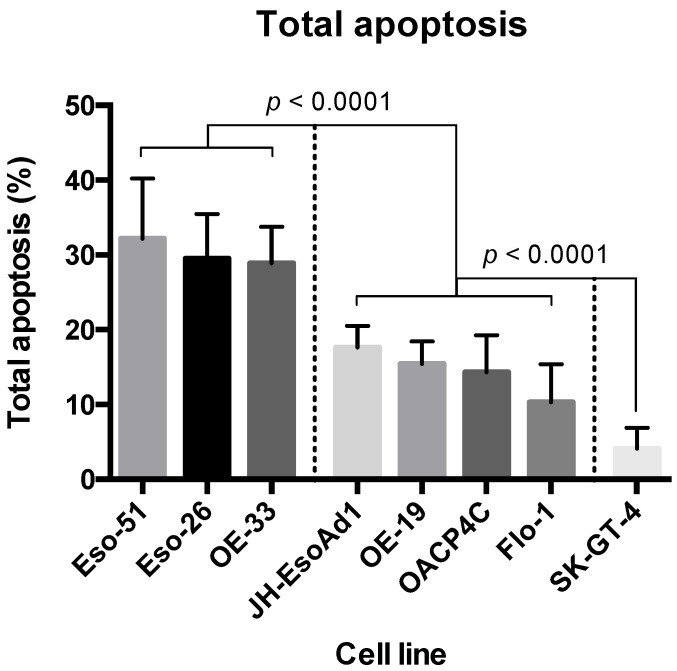
Summary of total apoptosis in eight oesophageal adenocarcinoma cell lines treated with 10 μM 4-hydroxytamoxifen. Results represent pooled data from three independent experiments each with three technical replicates (i.e., a total of nine replicates per treatment or control group). Columns and error bars represent means and standard deviations, respectively.

**Figure 4 cancers-14-01891-f004:**
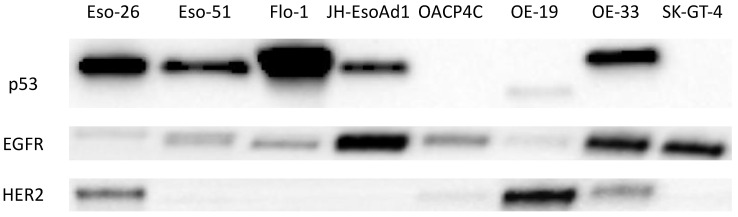
Western blots showing expression of p53, EGFR, and HER2 in oesophageal adenocarcinoma cell lines.

**Figure 5 cancers-14-01891-f005:**
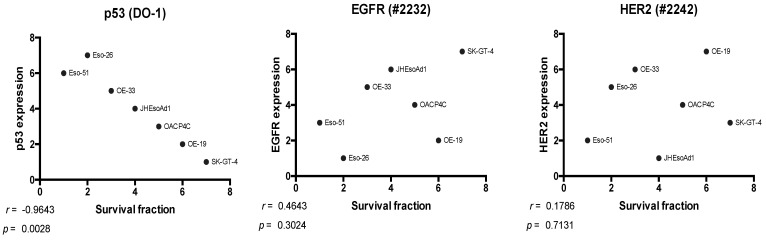
Scatterplots of rank showing relationships between survival fraction and the expression of p53, EGFR, and HER2. Spearman correlation. DO-1, anti-p53 primary antibody; #2232, anti-EGFR primary antibody; #2242, anti-HER2 primary antibody.

**Figure 6 cancers-14-01891-f006:**
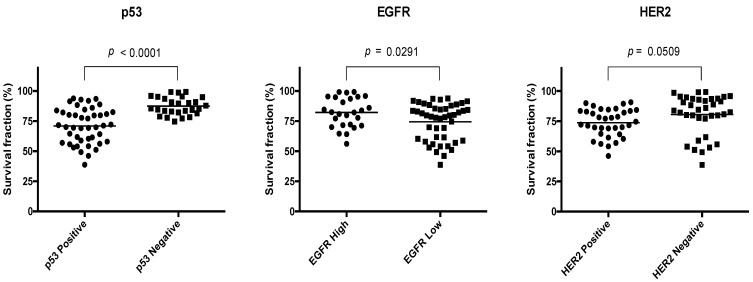
Relationship between baseline expression of p53, EGFR, and HER2, and survival fraction after treatment with 10 μM of 4-hydroxytamoxifen. Data points indicate pooled survival fractions from three independent experiments from cell lines treated in triplicate. Student’s T-test.

**Figure 7 cancers-14-01891-f007:**
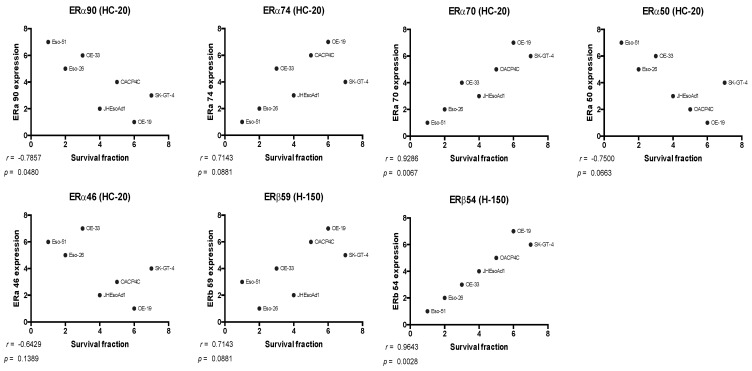
Scatterplots of rank showing relationships between survival fraction and the expression of ERα and ERβ species.

**Table 1 cancers-14-01891-t001:** Correlation matrix summarising relationships between ERα and ERβ species in oesophageal adenocarcinoma cells by Western blot. Shaded cells indicate significant correlations (*p* ≤ 0.05). ρ (rho), Spearman rank correlation coefficient.

**ERα74**	ρ = −0.619					
	*p* = 0.115					
**ERα70**	ρ = −0.738	ρ = 0.929				
	*p* = 0.046	*p* = 0.002				
**ERα50**	ρ = 0.929	ρ = −0.738	ρ = −0.762			
	*p* = 0.002	*p* = 0.046	*p* = 0.037			
**ERα46**	ρ = 0.952	ρ = −0.548	ρ = −0.619	ρ = 0.952		
	*p* = 0.001	*p* = 0.171	*p* = 0.115	*p* = 0.001		
**ERβ59**	ρ = −0.405	ρ = 0.690	ρ = 0.714	ρ = −0.548	ρ = −0.405	
	*p* = 0.327	*p* = 0.069	*p* = 0.058	*p* = 0.171	*p* = 0.327	
**ERβ54**	ρ = −0.857	ρ = 0.762	ρ = 0.881	ρ = −0.857	ρ = −0.786	ρ = 0.786
	*p* = 0.011	*p* = 0.037	*p* = 0.007	*p* = 0.011	*p* = 0.028	*p* = 0.028

## Data Availability

Not applicable.

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
