# Peer review of "Oestrogen Receptor Isoforms May Represent a Therapeutic Target in Oesophageal Adenocarcinoma"

_cancers, 2022, doi:10.3390/cancers14081891_

Round 1

Reviewer 1 Report

Steven L Due et al. analyze the significance of different estrogen receptor subtypes (and isoforms) in esophageal adenocarcinoma (EAC). For this purpose, they analyze different cell cultures using common techniques. They investigate the efficiency of therapeutic blockade with tamoxifen and correlate the results with expression levels of p53, EGFR and Her2/neu. Relevant in this context is that there is a positive association between mutant p53 and therapeutic sensitivity of tamoxifen, and in contrast, therapeutic resistance of tamoxifen and EGFR expressing tumors. The work is important in my view because it brings into focus the importance of hormone receptors in EAC. Many works are already known in gastric carcinoma, some with contradictory results, the more important is the approach of this topic in EAC. Some aspects, however, seem to me to be neglected or not addressed at all in the discussion. It is a long-known phenomenon that adenocarcinoma of the esophagus is a tumor disease that occurs predominantly in men (7-9:1). Accordingly, the sexes are distributed in the cell cultures analyzed (7:1). Sex-biological aspects in oncology are of great importance. This issue is not addressed at all. Response rate and expression correlations of cell cultures are one thing, making statements in humans is a completely different reality. Are there individual case reports in the medical literature from which we could conclude that tamoxifen/or other anti-hormonal treatments are at least partially effective in EAC (e.g. women with breast carcinoma under tamoxifen and co-existing EAC or men with EAC and co-existing prostate carcinoma under androgen blockade - even if the latter has nothing to do with tamoxifen it could still be an indication that anti-hormonal therapy has an effect on EAC). For years, the relevant literature has discussed disruption of the androgen receptor pathway as a reason for male dominance in EAC. This may be incorrect. However, since seven out of eight cell lines used are from males, at least this aspect should be considered in the discussion. If EGFR (EGFR pathway) induces resistance to tamoxifen, what happens if I block EGFR?  This should be mentioned as a future goal in the publication.

Author Response

Please see attached letter.

Reviewer 2 Report

Authors investigate the effect of tamoxifen  in EAC cell lines. Rationale is the previous identification of Estrogen receptors in EAC. 

Figure 1. I see a lot of bands on the Western Blot. The legend could use some more explanation. Why are there so many bands in A and why is only a selection shown in figure B? (its written in the text, but a figure should be self explanatory). What molecular weights were to be expected and did they differ? There is also a ? symbol behind the k of KDa. What were the negative and positive control used for HC-20 and H-150?

What ER species are present in normal mucosal biopsies? If any, what are the differences with EAC?

Tamoxifen treatment at 10 μM for 48 h led to differences. Why 10 uM , Why 48 hours. Were these concentrations and intervals previously determined? What is the effect of exposure in other (non-EAC) cell lines?

What are the characteristics of these cell lines. Do they differ in any other items? (e.g. driver mutation make-up, rate of growth?)

How do the cell lines compare to each other without tamoxifen exposure (eg do they differ in apoptosis without exposure?

Author Response

Please see the attached letter.
